# A temperature-induced metabolic shift in the emerging human pathogen *Photorhabdus asymbiotica*

Elena Lucy Carter,[1] Nicholas R. Waterfield,[1] Chrystala Constantinidou,[1,2] Mohammad Tauqeer Alam[3]

**ABSTRACT** *Photorhabdus* is a bacterial genus containing both insect and emerging human pathogens. Most insect-restricted species display temperature restriction, unable to grow above 34°C, while *Photorhabdus asymbiotica* can grow at 37°C to infect mammalian hosts and cause Photorhabdosis. Metabolic adaptations have been proposed to facilitate the survival of this pathogen at higher temperatures, yet the biological mechanisms underlying these are poorly understood. We have reconstructed an extensively manually curated genome-scale metabolic model of *P. asymbiotica* (iEC1073, BioModels ID MODEL2309110001), validated through *in silico* gene knockout and nutrient utilization experiments with an excellent agreement between experimental data and model predictions. Integration of iEC1073 with transcriptomics data obtained for *P. asymbiotica* at temperatures of 28°C and 37°C allowed the development of temperature-specific reconstructions representing metabolic adaptations the pathogen undergoes when shifting to a higher temperature in a mammalian compared to insect host. Analysis of these temperature-specific reconstructions reveals that nucleotide metabolism is enriched with predicted upregulated and downregulated reactions. iEC1073 could be used as a powerful tool to study the metabolism of *P. asymbiotica,* in different genetic or environmental conditions.

**IMPORTANCE** *Photorhabdus* bacterial species contain both human and insect pathogens, and most of these species cannot grow in higher temperatures. However, *Photorhabdus asymbiotica*, which infects both humans and insects, can grow in higher temperatures and undergoes metabolic adaptations at a temperature of 37°C compared to that of insect body temperature. Therefore, it is important to examine how this bacterial species can metabolically adapt to survive in higher temperatures. In this work, using a mathematical model, we have examined the metabolic shift that takes place when the bacteria switch from growth conditions in 28°C to 37°C. We show that *P. asymbiotica* potentially experiences predicted temperature-induced metabolic adaptations at 37°C predominantly clustered within the nucleotide metabolism pathway.

**KEYWORDS** metabolic modeling, flux balance analysis, genome scale model, *Photorhabdus*, stress adaptation

*P*hotorhabdus asymbiotica is an emerging human pathogen, recovered from clinical isolates of human wounds (1). Confirmed clinical cases of Photorhabdosis, typically presenting as a skin and soft tissue infection, though bacteraemia has also been reported, across the United States and Australia (2). However, the bacterium has not yet been included in rapid identification systems (3), so the impact and geographical spread of the pathogen could be more widespread than current data suggest. *Photorhabdus* are obligate symbionts of entomopathogenic nematodes, which are responsible for the transmission of *P. asymbiotica* to mammalian hosts. Photorhabdosis typically presents as wound infections, but several cases have also been reported for uninjured skin,

**Peer Reviewer** Maksim Zakhartsev, University of Stuttgart, Stuttgart, Germany

Address correspondence to Mohammad Tauqeer Alam, mtalam@uaeu.ac.ae.

The authors declare no conflict of interest.

See the funding table on p. 17.

suggesting that the nematode is able to burrow into the skin, delivering *Photorhabdus* subcutaneously (4).

Nearly all *Photorhabdus* species are insect-restricted, unable to establish disease in a mammalian host, while *P. asymbiotica* and *P. australis* are both emerging human pathogens. *Photorhabdus* species are typically unable to grow at temperatures exceeding 34°C (5, 6), introducing an absolute barrier to establishing infection in a mammalian host, partly due to the presence of the temperature restricting locus which restricts bacterial replication at temperatures above 34°C through a yet unknown mechanism (7). Transcriptomic studies of *P. asymbiotica* have revealed that the pathogen undergoes a dramatic metabolic shift at higher temperatures (8) to facilitate survival. At 37°C compared to 28°C, *P. asymbiotica* can utilize fewer carbon and nitrogen sources to support respiration (8), potentially evidencing a nutritional virulence strategy employed during mammalian infections to exploit available host nutrients for proliferation (9).

Metabolic alterations are pivotal in the *Photorhabdus* life cycle, facilitating the transition from a mutualistic association with the nematode to pathogenicity toward the insect, or potentially mammalian, host (10). *Photorhabdus* populations display phenotypic heterogeneity (11). The switch from a pathogenic to mutualistic life cycle is characterized by the production of antibiotics, pigments, and bioluminescence (10, 11). Similar adaptations are integral in the lifestyle decisions of other pathogenic bacteria including a switch from the metabolism of amino acids to glycerolipids in *Legionella pneumophila* for transmission (12) and the induction of triglyceride synthesis in *Mycobacterium tuberculosis* which contributes to drug resistance (13). Temperature is also closely associated with the regulation of metabolism, growth, and virulence in bacteria, particularly when experiencing changes in temperature between the environment and the host (14), or vice versa. For example, exposure to low temperatures suppresses the growth of *Salmonella* (14) while high temperatures induce lethal alterations in the cytoplasm and cell envelope of such species (14, 15).

This study, which uses a combination of a newly reconstructed genome-scale metabolic model (GEM) and published transcriptome data, aims to understand the metabolic changes that occur in *P. asymbiotica* following a temperature change inducing a metabolic shift. GEMs computationally describe the entire set of metabolic reactions existing within an organism which can be analyzed using constraint-based analysis approaches (16, 17). GEMs have been used as a major tool to study an organism's metabolism at a systems-level and generate organism-specific biological predictions (18, 19). Current applications of such metabolic reconstructions include the modeling of microbial communities (20–23), metabolic engineering (24, 25), identification of drug targets (26–28), and use in industry. The integration of transcriptomic data sets allows the reconstruction of context-specific models providing an overview of metabolism between various conditions (29) including different media (30), metabolic backgrounds (30, 31), environments (32), and tissues (33–35).

The creation of temperature-specific reconstructions for *P. asymbiotica* metabolism in this study reveals the metabolic adaptations the pathogen undergoes at higher temperatures. Our results reveal that nucleotide metabolism, amongst other metabolic pathways, is enriched with predicted upregulated and downregulated reactions, particularly in reference to the biosynthesis of precursors required for DNA replication.

## RESULTS

### Genome-scale metabolic model of *P. asymbiotica* ATCC43949 (iEC1073)

The genome-scale metabolic model of *P. asymbiotica* ATCC43949 ($Pa^{ATCC43949}$) was reconstructed following a comprehensive workflow (17) (Fig. 1a), described in Materials and Methods in detail. Briefly, using the genome sequence (1) (accession number NC_012962.1), a draft reconstruction was produced by the Model SEED platform (36). This *Photorhabdus* strain was selected for this study as it grows at both 28°C and 37°C, and a transcriptomics data set of the strain growing at both these temperatures is published (8); such data are not widely available for *Photorhabdus* species. In addition, *P.*

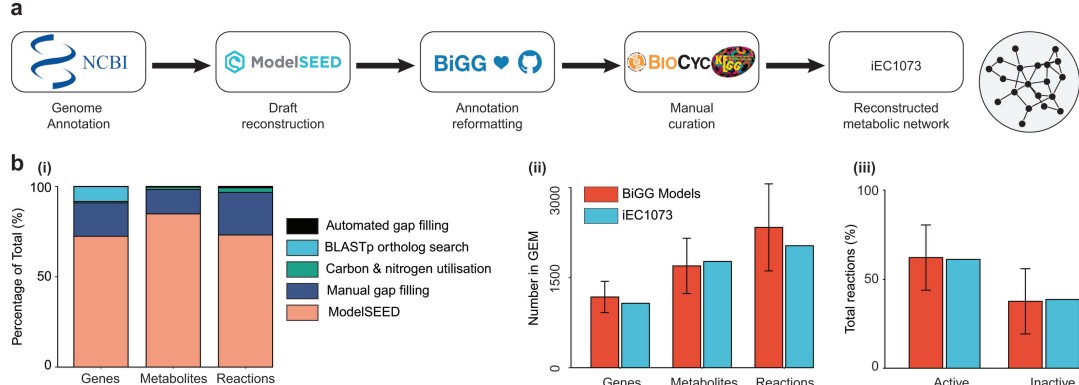

**FIG 1** Metabolic reconstruction of *Photorhabdus asymbiotica* (iEC1073). (a) Schematic for reconstructing the metabolic model of *P. asymbiotica*. It includes the generation of a draft reconstruction using the genome sequence by the Model SEED, integration of BiGG identifiers, and extensive manual curation. (b) iEC1073 contains 1,073 genes; 1,770 metabolites; and 2,033 reactions. These genes, metabolites, and reactions were added throughout the reconstruction process at different stages of curation (i). The number of genes, metabolites, and reactions in iEC1073 is comparable with other high quality, thoroughly curated prokaryotic GEMs available on the BiGG database (37) (ii). The proportion of active and inactive reactions in iEC1073, in unconstrained conditions, is comparable with those for five selected published prokaryotic GEMs (38–42) (iii).

*asymbiotica* is of interest as a known human pathogenic species of the genus, capable of causing Photorhabdosis and more widespread infection, such as endocarditis (1).

The obtained draft model contained 1,486 reactions; 1,501 metabolites; and 777 genes in total; however, it was unable to produce biomass *in silico* under a defined minimal media condition (8). Therefore, the draft model was further curated (Fig. 1a). For curation and standardization, however, the draft model was first reformatted to the BiGG database nomenclature (37). For model curation, a combination of several approaches was employed including automated gap filling, a gene ortholog search, and manual gap filling by consulting various biochemical databases such as MetaNetX (43) and KEGG (44), alongside experimental observations (Fig. 1a). This gap filling was performed in addition to the automated gap filling employed by the Model SEED platform during automated network reconstruction (36).

A total of 87 missing orthologous metabolic genes were identified following the reciprocal best BLASTp hits (45) against five closely related prokaryotes including *Bacillus subtilis*, *Escherichia coli,* and *Yersinia pestis*. These ortholog genes (File S1) were added to the reconstruction (Fig. 1b, panel i) to improve the gene-protein reaction associations for existing reactions in the reconstruction. In addition, all annotated KEGG pathways were checked, and missing metabolic genes, associated metabolites, and associated reactions were added. Care was taken during these curation efforts to minimize introducing dead-end metabolites (DEMs) into the network; DEMs are those which are only consumed or produced in a network and, therefore, block the flux of downstream reactions. Such metabolites were only added to the reconstruction when involved in a reaction associated with a gene not yet accounted for in the reconstruction.

*Photorhabdus* are bioluminescent, and this bioluminescence is conferred by enzymes encoded by the *lux* operon (46). This operon is conserved across the entire *Photorhabdus* genus, suggesting it is critical to the survival of the pathogen, while known to be non-essential *in vitro*, the exact biological advantage remains to be elucidated. Genes *luxA* and *luxB* encode the alpha and beta subunits of the luciferase enzyme (46) which catalyzes the reaction resulting in light emission. Genes *luxC*, *luxD,* and *luxE* encode enzymes making up the fatty acid reductase complex catalyzing the biosynthesis of long-chain aldehydes (46, 47), a substrate in the photon-producing reaction. Furthermore, *Photorhabdus* species also produce various antimicrobial compounds during their life cycle, most notably stilbene compounds including 2-isopropyl-5-[(E)-2-phenylvinyl]benzene-1,3-diol and 2-ethyl-5-[(E)-2-phenylvinyl]benzene-1,3-diol which are thought to play a crucial role during the establishment of nematode mutualism (48).

Reactions involved in stilbene biosynthesis (48) and those conferring bioluminescence were added to the network (File S2). To encapsulate the importance of the bioluminescence pathway in the *Photorhabdus* life cycle, the photon metabolite (photon) was also added as a reactant in the biomass reaction in the reconstruction.

In addition to gap filling, during manual curation, the identification of DEMs was also necessary. As previously mentioned, the presence of a DEM in a pathway blocks downstream flux of associated reactions. While the majority of these DEMs were corrected for during gap filling, for the remaining compounds, the addition of an associated transport and exchange reaction could address the issue, allowing the consumption of such metabolites by removing them from the *in silico* mtabolic network. During manual curation, a total of 199 genes, 483 reactions, and 239 metabolites (Fig. 1b, panel i) were added, while three reactions were removed to be replaced with the correctly annotated reactions instead. The majority of these additions were a result of manual gap filling, but 17 reactions, 7 metabolites, and 1 gene were added following automated gap filling (Fig. 1b, panel i). The reactions identified through automated gap filling consisted of transport or exchange reactions, and the corresponding metabolites were the extracellular versions of cytosolic ones already present in the reconstruction. The remaining genes, metabolites, and reactions added to the network were added at a later stage during the model validation process while comparing with phenotype microarray data for the utilization of various carbon and nitrogen sources (mentioned in the results section describing the integration of phenotype microarray data). While this is discussed in more detail in the following section, a total of 9 genes, 23 metabolites, and 50 reactions were added (Fig. 1b, panel i) to finalize the network. So, in total, 296 genes, 269 metabolites, and 547 reactions were added to create the final metabolic reconstruction for $Pa^{ATCC43949}$ (iEC1073, BioModels ID MODEL2309110001) containing 1,073 genes; 1,770 metabolites; and 2,033 reactions (File S2).

Aside from exchange reactions, which do not have a GPR association as standard, iEC1073 currently contains 91 reactions (File S2) without an associated gene (4.5% of total reactions). Twenty-nine of these unannotated reactions are spontaneous reactions, as denoted in the BiGG database, and predominantly consist of transport reactions. The biomass reaction is also unannotated. The remaining 61 reactions (3% of total reactions) are required in the network to fill gaps in metabolic pathways, but no candidate gene could be identified in *P. asymbiotica*. This proportion of unannotated reactions in prokaryotic genome-scale models is common, for example, 4% of all reactions in iML1515 for *E. coli* (38) are unannotated. Unannotated metabolic reactions represent reactions which are predicted to exist in a given organism to complete various pathways but do not yet have a known enzyme encoded by the organism to facilitate the reaction.

The number of genes, metabolites, and reactions in iEC1073 is comparable with other prokaryotic GEMs available on the BiGG models platform (37) (Fig. 1b, panel ii). Of the 2,033 reactions in the network, 1,745 are metabolic; 165 are transport; and the remaining 123 are exchange reactions (File S2). iEC1073 produces biomass *in silico* in both unconstrained, all exchange reactions open, and constrained, simulating a defined minimal media (8), conditions at a rate of 135.4065 mmol/gDW/h and 0.6530 mmol/gDW/h, respectively.

In unconstrained conditions, 787 (39%) reactions were inactive, while 1,246 (61%) were active and carrying flux. This distribution was compared to the published metabolic reconstructions for *B. subtilis* (iYO844) (39), *E. coli* (iML1515) (38), *H. pylori* (iIT341) (40), *M. tuberculosis* (iEK1008) (41), and *Y. pestis* (iPC815) (42) and is comparable (Fig. 1b, panel iii). As expected, in constrained conditions, the number of inactive reactions in the metabolic reconstruction for *P. asymbiotica* increased to 911 (45%) (Fig. S1). The final model could be used to mimic different genetic and environmental perturbations, including nutrient supplementation, stress tolerance, or adapting to different hosts, to understand the metabolic re-arrangement in such conditions.

## Validation of iEC1073 through comparison of orthologous gene essentiality

The network reconstruction for $Pa^{ATCC43949}$ (iEC1073, BioModels ID MODEL2309110001) was validated through comparison with experimental results to ensure that it was biologically representative of the organism, particularly, so it can be used as a predictive phenotypic tool or as a platform to start investigating the metabolism of *P. asymbiotica*. The iEC1073 reconstruction was validated through both gene essentiality analysis, and nutrient source utilization experiments, which have both been shown to validate the predictive accuracy of metabolic reconstructions (49–52).

*In silico* single gene knockout experiments were performed in unconstrained media conditions (similar to rich media), with all exchange reactions open (Fig. 2a). In total, out of the 1,073 genes in the reconstruction, 194 (18.1%) were predicted to be essential (File S3). Unfortunately, no experimental gene essentiality data exists for *P. asymbiotica*, or, indeed, any *Photorhabdus* species, so these predicted gene essentiality statuses were instead compared to the experimental results for orthologous genes in closely related organisms including *B. subtilis, E. coli, H. pylori, M. tuberculosis,* and *Y. pestis,* obtained from the Online Gene Essentiality (OGEE) database (53) (https://v3.ogee.info/). Remarkably, 62%–86% of the iEC1073 predicted essential genes have orthologs with the same essentiality status in these species including the maximum 86% in *E. coli,* which is one of the most highly curated models and phylogenetically one of the closest related species to *Photorhabdus,* and 81% in *B. subtilis* (Fig. 2b). The experimental data for gene essentiality in these species was also compared against its own predicted gene essentiality in its published metabolic reconstruction (38–42). Compared to the results obtained for the corresponding experimental results, these GEMs were 82%, 91%, 52%, 87%, and 68% concordant for *B. subtilis*, *E. coli*, *H. pylori*, *M. tuberculosis,* and *Y. pestis*, respectively (Fig. 2b). These results confirm that no metabolic reconstruction is entirely representative of the biological activity of the species it represents. Interestingly, *H. pylori*,

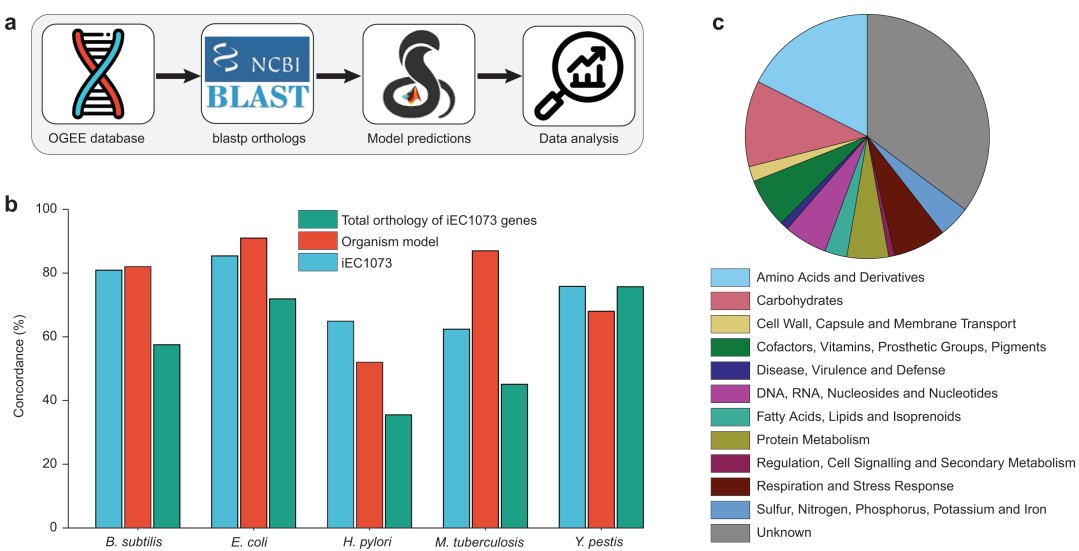

**FIG 2** Comparing the predicted essential genes with experimental data. (a) Schematic of *in silico* single gene knockouts of iEC1073 compared to experimental data for validation. Experimental gene essentiality data were obtained from the online gene essentiality (OGEE) database (53) for five comparative organisms (*B. subtilis, E. coli, H. pylori, M. tuberculosis,* and *Y. pestis*). *In silico* single-gene knockout experiments were conducted for iEC1073 and predicted essentiality status compared with orthologous genes in comparison organisms. (b) Concordance of gene essentiality in iEC1073 compared to orthologous genes in comparative organisms (cyan), concordance of the published metabolic reconstruction of these organisms with their own experimental data (red), and the total number of orthologous genes compared between iEC1073 and the comparative organism (green). iEC1073 is 81%, 86%, 65%, 62%, and 76% concordant with *B. subtilis, E. coli, H. pylori, M. tuberculosis,* and *Y. pestis,* respectively. Quite similarly, the published metabolic network reconstructions for *B. subtilis, E. coli, H. pylori, M. tuberculosis,* and *Y. pestis* are 82%, 91%, 52%, 87%, and 68% concordant with experimental data sets of the same species, respectively. (c) Pathway distribution of essential genes in iEC1073 concordant with the essentiality status of *E. coli* orthologs (53). The majority of these genes have an unannotated role, while the remaining genes are evenly distributed among the major annotated metabolic categories.

which had concordance of only 65% between experimentally observed essential genes and iEC1073 predicted essential genes, is also the lowest against its own predicted essential genes (Fig. 2b). This could potentially be explained by the highly variable nature of the *H. pylori* genome, with mutation rates described as more than 10 times higher than those observed in *E. coli* (54).

However, *E. coli* has the maximum concordance with its own model prediction as well as with the iEC1073 predicted essential genes. The high similarity between the model predictions and experimental observations indicates the quality of the model, which can be used as a predictive tool. Furthermore, analysis of the metabolic pathways involving concordant genes, when compared to *E. coli*, revealed that most analyzed genes do not have an annotated function and the remainder were relatively evenly distributed among different metabolic pathways (Fig. 2c). Furthermore, the fact that the discordant genes are not primarily associated with one or a couple of primary metabolic pathways (Fig. S2) suggests that there is no area of the network reconstruction which still requires major curation.

## Validation of iEC1073 through integration of phenotype microarray data

Phenotype microarray data have been collected for *P. asymbiotica* at both 28°C and 37°C (8) and was used to validate iEC1073 based on the utilization of different carbon and nitrogen sources. This microarray experiment was integral in the identification of the temperature-induced metabolic shift observed in *P. asymbiotica* with the pathogen being only able to utilize a minimal selection of nutrients to support respiration at 37°C compared to 28°C (8). Experimental results describing the metabolic activity of *P. asymbiotica* at a temperature of 28°C were used for comparison with predictions made by iEC1073.

It should be noted that only the limited number of 190 carbon sources were tested as part of this microarray experiment (8), 84 of which exist in the iEC1073 metabolic model and that 36 of these have a corresponding transport and exchange reaction facilitating uptake from the *in silico* extracellular environment. Additionally, 95 nitrogen sources were tested (8), 62 of which exist in the iEC1073 metabolic model and 21 of these have a corresponding transport and exchange reaction. Based on the experimental data, 15 additional carbon sources and 8 additional nitrogen sources, which existed in the metabolic reconstruction but without a transport or exchange reaction, were identified as being able to support the growth of *P. asymbiotica* experimentally. Transport and exchange reactions were added for these metabolites alongside an extracellular version of these metabolites (Fig. 1b, panel i) to allow for uptake of the nutrient source.

A total of 51 carbon sources and 29 nitrogen sources were tested to identify whether these nutrients could support the production of biomass *in silico*, simulating media conditions used during the phenotype microarray experiments (8) (Table 1). False negative results refer to nutrients which can support the respiration of the bacterium experimentally, but which cannot be utilized by the network to support the production of biomass *in silico*, and, therefore, indicate gaps in the reconstruction. Gaps addressed by this analysis include those surrounding maltotriose metabolism, for example, which was corrected by adding a maltodextrin glucosidase reaction, catalyzing the conversion of maltotriose to glucose and maltose (R_MLTG1). However, some of these discrepancies could not be fixed at this stage due to a lack of suitable annotated genes catalyzing an appropriate reaction, such as is the case for the utilization of inositol, tyramine, and the dipeptide Ala-Leu.

Of the 51 carbon sources tested (File S4), 72.5% are concordant between the experimental phenotype microarray results and the model predictions (Fig. 3a, panel i). In 37 conditions, both iEC1073 and the experimental data are in agreement. Of the remaining 14 conditions, for only three carbon sources, M-inositol, tyramine, and succinate, is the reconstruction unable to produce biomass despite the organism being shown to respire in the presence of these nutrients. The remaining 11 discordant conditions consist of false positive results, whereby the model predicts

**TABLE 1** Defined minimal media supporting *Photorhabdus* growth experimentally[a]

| Modified M9 salts | Uptake rate (mmol/gDw/h) |
| --- | --- |
| Hydrogen (M_h_c) | −1,000 |
| Sodium (M_na1_c) | −1,000 |
| Water (M_h2o_c) | −1,000 |
| Magnesium (M_mg2_c) | −1,000 |
| Potassium (M_k_c) | −1,000 |
| Phosphate (M_pi_c) | −1,000 |
| Sulphate (M_so4_c) | −1,000 |
| Oxygen (M_o2_c) | −2 |
| Chlorine (M_cl2_c) | −1,000 |
| Calcium (M_ca2_c) | −1,000 |
| Casein/casamino acids | |
| Glycine (M_gly_c) | −0.22 |
| Proline (M_pro__L_c) | −1.56 |
| Glutamate (M_glu__L_c) | −0.85 |
| Arginine (M_arg__L_c) | −0.18 |
| Serine (M_ser__L_c) | −0.71 |
| Alanine (M_ala__L_c) | −0.40 |
| Lysine (M_lys__L_c) | −0.54 |
| Tryptophan (M_trp__L_c) | −0.04 |
| Tyrosine (M_tyr__L_c) | −0.18 |
| Phenylalanine (M_phe__L_c) | −0.40 |
| Histidine (M_his__L_c) | −0.22 |
| Isoleucine (M_ile__L_c) | −0.49 |
| Methionine (M_met__L_c) | −0.31 |
| Aspartate (M_asp__L_c) | −0.18 |
| Asparagine (M_asn__L_c) | −0.22 |
| Glutamine (M_gln__L_c) | −0.89 |
| Threonine (M_thr__L_c) | −0.40 |
| Carbon source | |
| Mannose (M_man_c) | −10 |
| Trace elements | |
| Cobalt (M_cb2_c) | −1,000 |
| Iron (M_fe2_c) | −1,000 |
| Zinc (M_zn2_c) | −1,000 |
| Copper (M_cu2_c) | −1,000 |
| Manganese (M_mn2_c) | −1,000 |

[a]Adapted from Mulley et al. (8). Biomass production simulated *in silico* utilizing the COBRA toolbox (55) in both constrained and unconstrained conditions.

biomass production despite a lack of observed metabolic activity experimentally. We believe these results could potentially be explained by an experimental respiration rate occurring below a certain detection level. Of the 51 carbon sources tested, 39 increased metabolic activity and 12 reduced this activity experimentally (Fig. 3a, panel ii), while 39 increased biomass production and 12 did not change the rate of biomass production by iEC1073 (Fig. 3a, panel iii). Interestingly, *N*-acetyl-ᴅ-glucosamine (GlcNAc) best supported experimental respiration and increased biomass production the second best in iEC1073 (Fig. 3a, panels ii and iii). The dipeptide ʟ-Alanyl-Glycine (Ala-Gly) best supported biomass production in iEC1073 (Fig. 3a, panel iii).

For the 29 nitrogen sources tested (File S4), 62% of the model predictions are concordant with the experimental phenotype microarray data (Fig. 3b, panel i). Of these 29 tested conditions, 62% are concordant between the experimental data and the model predictions. Of the 18 concordant conditions, for 10 nitrogen sources, the organism was shown to grow *in silico* and respire in the lab, and for the remaining 8 of these

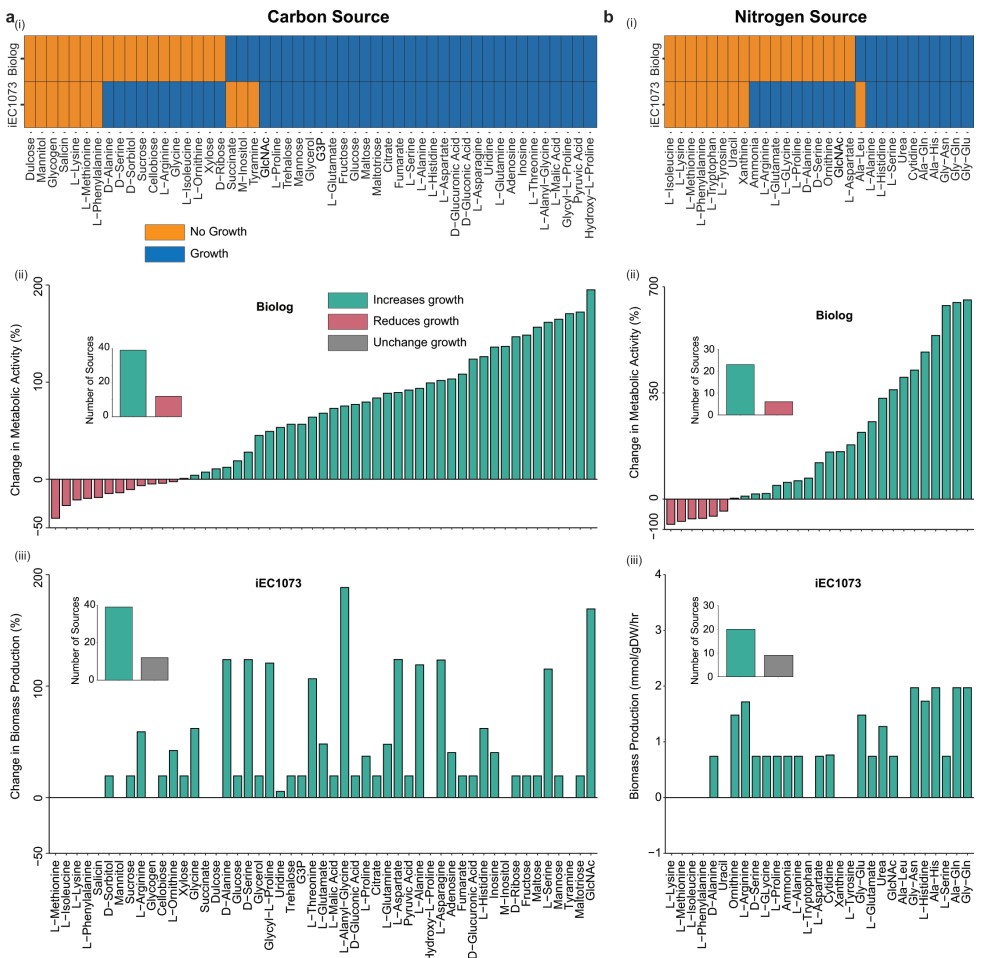

**FIG 3** Model validation using experimental phenotype microarray data (8). *In silico* carbon and nitrogen utilization predictions were used to validate iEC1073 with experimental phenotype microarray data. (a) 51 carbon sources were tested in iEC1073. A total of 37 (72.5%) of these carbon sources are concordant with the experimental data (i). Of the carbon sources discordant between the reconstruction and experimental data, for only three can the reconstruction not produce biomass despite respiration observed experimentally (i). Of the 51 carbon sources tested, 39 sources increased metabolic activity and 12 decreased this activity experimentally (ii), and 39 sources increased biomass production by iEC1073 while 12 could not support an increase in biomass production (iii). *N*-Acetyl-D-glucosamine (GlcNAc), an amide derivative of glucose, best supported metabolic activity of the organism in the lab and also increased *in silico* growth, though the dipeptide Ala-Gly most increased biomass production in iEC1073 (ii, iii). (b) Twenty-nine nitrogen sources were tested in iEC1073. A total of 18 (62%) of these nitrogen sources are concordant with the experimental data (i). For only one of the discordant nitrogen sources, the dipeptide Ala-Leu was the reconstruction unable to produce biomass despite respiration observed experimentally (i). Of the 29 nitrogen sources tested, 23 sources increased metabolic activity and 6 decreased this activity experimentally (ii), and 20 sources increased biomass production by iEC1073 while 9 could not support an increase in biomass production (iii). The dipeptide Gly-Gln most increases experimental metabolic activity and this dipeptide, along with Gly-Asn, Ala-His, and Gly-Gln, best support biomass production by the reconstruction (ii, iii).

concordant conditions, no activity was observed *in silico* or experimentally. The remaining 11 tested conditions are discordant between iEC1073 and the experimental phenotype microarray data. For only 1 nitrogen source, the dipeptide L-alanyl-leucine (Ala-Leu) is the reconstruction unable to produce biomass despite the organism being shown to metabolize this compound experimentally. Of the 29 nitrogen sources tested, 23 increased metabolic activity and 6 reduced this activity experimentally (Fig. 3b, panel ii) while 20 increased biomass production and 9 did not change the rate of biomass production by iEC1073 (Fig. 3b, panel iii). Interestingly, dipeptides best support experimental respiration and biomass production by iEC1073 (Fig. 3b, panels ii and iii).

L-Glycine-glutamine (Gly-Gln) best supports metabolic activity, while this dipeptide along with L-glycine-asparagine (Gly-Asn), L-alanine-histidine (Ala-His), and L-Alanine-Glutamine all result in the same maximum increase in biomass production in the reconstruction (Fig. 3b, panels ii and iii). The fact that both of these nitrogen sources are all dipeptides, and that a dipeptide also best supports respiration for the tested carbon sources, suggests that the metabolic reactions associated with the metabolism of dipeptides are highly curated in the metabolic reconstruction and are representative of the growth phenotype of *P. asymbiotica*.

Though the *in silico* growth rate of iEC1073 and the individual nutrient source uptake rates differ between the two set lower bounds of −10 mmol/gDw/h and −1,000 mmol/gDw/h for the appropriate exchange reaction, the general consensus remains the same (File S4).

## Integration of transcriptomics data reveals temperature-induced metabolic differences in *P. asymbiotica*

Transcriptomics data obtained for *P. asymbiotica* at a temperature of 28°C and 37°C (8) were integrated with the validated reconstruction obtained for *P. asymbiotica* to create temperature-specific networks detailing the metabolic alterations the organism undergoes at higher temperatures and, therefore, potentially in a mammalian compared to insect host. This data set describes transcriptomics changes induced by a temperature of 37°C compared to 28°C in *P. asymbiotica* growing in planktonic cultures; no other variables were studied (8).

Flux variability analysis (FVA) was performed, employing the fluxVariability algorithm in the COBRA Toolbox (55) to predict the lower and upper flux bound of each reaction in the network to support maximum biomass production in unconstrained conditions, simulating growth in LB. The predicted reaction flux bounds were considered for the 28°C temperature-specific reconstruction. To adjust this reconstruction to represent metabolism at 37°C, log2fold gene expression changes at a temperature of 37°C compared to 28°C (8) were mapped to each gene in the reconstruction, utilizing the findUsedGeneLevels and selectGenefromGPR functions in the COBRA Toolbox (55) to account for the gene-protein reaction association (File S5).

Next, the predicted flux bounds were adjusted to obtain the 37°C temperature-specific reconstruction. Of the total 1,819 reactions associated with 1,073 genes (91 reactions have no associated gene and 123 reactions represent the exchange of metabolites), the bounds of 348 reactions (19.1%) could not be altered as the resulting bounds rendered *in silico* biomass production infeasible. The flux bounds of these reactions were unchanged and kept as those in the 28°C temperature-specific reconstruction, as defined by FVA. For both temperature-specific models, the flux of each reaction in the network was predicted, setting the biomass reaction as the objective function, to study the temperature-induced metabolic changes occurring in the network (File S5).

Asides from metabolic changes, experimentally we observe different growth phenotypes of *P. asymbiotica* at these two temperatures, with the organism reaching an exponential phase of growth faster at a temperature of 37°C when grown in LB over 48 h while at 28°C, the organism displays a longer lag phase and higher maximum optical density (Fig. 4a; File S5), indicating that temperature influences the physiology of this pathogen outside of just metabolism.

Differences between the flux of reactions and the production of metabolites between the two temperature-specific models were analyzed (File S5). In both temperature-specific models, only 574 reactions carry fluxes (>1e−4), indicating the core set of metabolic reactions essential for growth. Of these core flux-carrying reactions, the flux of 394 reactions (68.6%) remains unchanged between the two conditions. However, in the 37°C model, a total of 100 reactions (17.4% of the total non-zero flux-carrying reactions) were predicted to be upregulated (Fig. 4b). In addition, a total of 80 reactions (13.9% of the total non-zero flux-carrying reactions) were predicted to be downregulated (Fig. 4b).

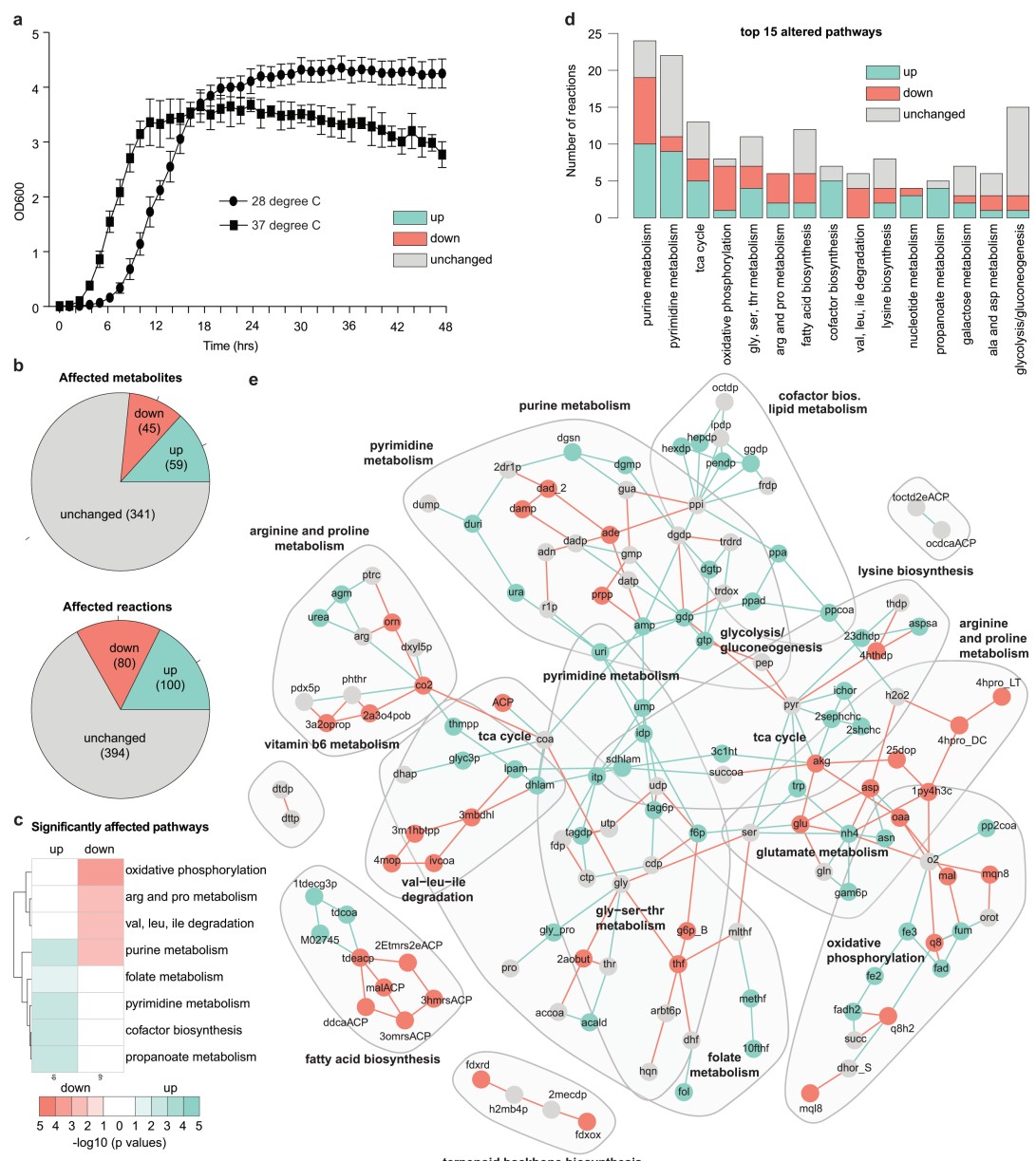

**FIG 4** Integration of transcriptomics data creates temperature-dependent reconstructions. The integration of gene expression data (8) for growth at 37°C and 28°C created temperature-dependent reconstructions detailing metabolic changes facilitating growth at higher temperatures. (a) *P. asymbiotica* displays different growth phenotypes depending on temperature, at 37°C the organism reaches an exponential phase of growth quicker in liquid cultures than when grown at 28°C. (b) Of the 1,077 total metabolites in the organism, 445 have a non-zero net production in at least one of the temperature-dependent reconstructions. Of these metabolites, flux through 341 (76.6%) remains unchanged, 59 (13.3%) are upregulated and 45 (10.1%) are downregulated at 37°C compared to 28°C. Of the 2,033 total reactions in the organism, 574 reactions carry non-zero flux in at least one of the temperature-dependent reconstructions. Of these reactions, flux through 394 (68.6%) remains unchanged, 100 (17.4%) are upregulated and 80 (13.9%) are downregulated at 37°C compared to 28°C. (c) The majority of the active reactions involved in folate metabolism, pyrimidine metabolism, cofactor biosynthesis, and the metabolism of propanoate are upregulated at 37°C compared to 28°C. In contrast, the majority of the active reactions associated with oxidative phosphorylation and the metabolism of certain amino acids are downregulated. Reactions involved in purine metabolism appear to be both upregulated and downregulated at 37°C compared to 28°C. (d) Most metabolic pathways predicted to be altered in response to temperature are associated with reactions that are upregulated, downregulated or experience no change in flux at 37°C compared to 28°C. (e) The predicted metabolic network containing upregulated and downregulated metabolites and reactions at 37°C compared to 28°C. Particular reactions and metabolites cluster into particular subsystems or pathways, but the temperature-induced metabolic changes span a wide range of functions.

Also, out of the total non-zero flux-carrying reactions, 544 reactions carry flux in both temperatures, 22 reactions are switched on in the 37°C model, being active at this

temperature but inactive at 28°C, and, vice versa, 8 reactions are switched off. The majority of these switched-on reactions are associated with amino acid metabolism but other pathways including lipid metabolism, the metabolism of cofactors, and pyrimidine metabolism are also affected.

Moreover, flux alterations in the 37°C temperature-specific reconstruction also result in differences in the net production of metabolites in the network. In both temperature-specific networks, a total of 445 metabolites have non-zero production flux, indicating the subgroup of metabolites associated with those core metabolic reactions essential for growth. Of these core metabolites, the net production of 341 metabolites (76.6%) remains unchanged between the two conditions. However, in the 37°C temperature-specific reconstruction, the net production of a total of 59 metabolites (13.3% of the total non-zero flux-carrying metabolites) was predicted to be upregulated (flux increase >5%) (Fig. 4b). In contrast, only 45 metabolites (10.1% of the total non-zero flux-carrying metabolites) were predicted to be downregulated (flux decrease > 5%) at 37°C compared to 28°C (Fig. 4b). Furthermore, 433 metabolites are produced in both models, and an additional 8 metabolites are predicted to be produced by the model at 37°C while not produced at all at 28°C, while the opposite is true for 4 metabolites (File S5).

Reactions and metabolites present in the reconstruction are associated with specific subsystems (File S2) detailing the different roles each has in metabolism. The pathways predicted to be upregulated at 37°C compared to 28°C include pyrimidine metabolism, propanoate metabolism, biosynthesis of cofactors, and folate metabolism (Fig. 4c and d). In contrast, oxidative phosphorylation and the metabolism of certain amino acids are predicted to be downregulated at 37°C (Fig. 4c and d). At 37°C, reactions associated with purine metabolism are both upregulated and downregulated (Fig. 4c and d).

*In silico* single-gene knockouts were repeated with the temperature-specific models and compared to the wild-type reconstruction (File S2). For the 28°C reconstruction, the number of predicted essential genes increased from 194 in the wild-type model to 291, and this number increased further to 311 for a temperature of 37°C (Fig. S3). Across all models, the essentiality status remains the same for 955 genes (89%); 194 genes are predicted to be essential in all conditions and 761 genes non-essential ( File S3).

To visualize the distribution of all predicted upregulated and downregulated metabolites and reactions in iEC1073 at 37°C compared to 28°C, a network was created (Fig. 4e). This network details that the temperature-induced metabolic changes experienced by *P. asymbiotica* at higher temperatures are primarily associated with the metabolism of nucleotides, the biosynthesis of cofactors, and energy metabolism. Temperature also appears to impact several other metabolic subsystems and pathways too, including terpenoid backbone synthesis and fatty acid metabolism, suggesting that these varied metabolic changes may all contribute to the ability of this organism to survive at higher temperatures.

## DISCUSSION

iEC1073 details a fully reconstructed and validated genome-scale metabolic network of the emerging human pathogen *Photorhabdus asymbiotica*, the first metabolic reconstruction for this species and for the *Photorhabdus* genus in general. Alterations to metabolism are vital during the life cycle of this genus, facilitating a change from a mutualistic relationship with the nematode symbiont to a pathogenic one directed at the insect host (10). This metabolic reconstruction will provide a platform for further studying the metabolic alterations contributing to the survival of this pathogen in differential environmental conditions, including differing temperatures and hosts.

The metabolic reconstruction is highly curated, incorporating knowledge from the literature and several biochemical databases, ensuring the network best describes the known metabolic capabilities of the organism to date. Furthermore, iEC1073 has also been successfully validated through the integration of experimental data, including phenotype microarray results detailing carbon and nitrogen nutrient utilization (8). Considering the lack of experimental gene essentiality data for *P. asymbiotica*, essentiality

data for five closely related and well-studied bacterial species was compared with *in silico* gene knockout experiments. The high levels of concordance with other bacterial species are particularly encouraging. Using orthologs for analysis, it is expected to identify genes which are both unique to *Photorhabdus* and discordant between species. Predictions of the utilization of different carbon and nitrogen sources for biomass production *in silico* are highly concordant with experimental results obtained for *P. asymbiotica*, further emphasizing the importance of curation efforts to ensure a reconstruction is representative of the experimentally observed phenotype. For the very few nutrients which could support respiration experimentally but were unable to facilitate biomass production by iEC1073, it is likely that the gene(s) facilitating such metabolic reactions is currently unannotated, highlighting the importance of updating metabolic reconstructions with the most recent genome annotation data to further fill any gaps present in the network. The experimentally validated model could be used as a tool to study the metabolism of the species in different genetic or environmental perturbations.

It has previously been described that *P. asymbiotica* undergoes a metabolic shift when grown planktonically at a temperature of 37°C compared to 28°C, an adaptation ascribed to a nutritional virulence strategy (8).

At this higher temperature, this analysis predicted that propanoate metabolism, cofactor biosynthesis, pyrimidine metabolism and folate metabolism are upregulated whilst oxidative phosphorylation and the metabolism of various amino acids are downregulated (Fig. 4c). Analyzing the metabolic alterations in the reconstruction induced by higher temperatures reveals the predicted upregulation of reactions in the pathways of propanoate metabolism, cofactor biosynthesis, pyrimidine metabolism and folate metabolism (Fig. 4e ). In contrast, oxidative phosphorylation and the metabolism of various amino acids are predicted to be downregulated (Fig. 4e).

A temperature-induced stress response has also been predicted in *Y. pestis* through constraint-based analysis approaches. Similar to *Photorhabdus*, the *Yersinia* life cycle involves an insect and mammalian host. The temperature shift experienced upon entry into the mammalian host has been predicted to downregulate oxidative phosphorylation (56), as is predicted for *P. asymbiotica* (Fig. 4c). This could potentially protect against oxidative stress encountered in phagocytic cells of the host (56). We also predict an increased flux for the galactose metabolic pathway (Fig. 4e), which has previously been shown to act as resistance against oxidative stress. Galactose can react with amines to form advanced glycation end products which upon receptor binding form reactive oxygen species (57) and, therefore, metabolizing this sugar limits the production of such free radicals. Upregulated galactose metabolism is also associated with biofilm formation in *B. subtilis* (58).

Metabolic changes have also been observed transcriptionally in *E. coli* following a temperature shift. Unlike *P. asymbiotica* and *Y. pestis*, *E. coli* is routinely grown at 37°C in laboratory conditions. At lower temperatures, *E. coli* upregulates genes associated with the TCA cycle (59). In *P. asymbiotica*, the majority of reactions associated with the TCA cycle are unaffected by temperature (Fig. 4d) though those that are affected are mostly upregulated at higher temperatures, with 5 reactions being upregulated at 37°C and 3 downregulated at this temperature (Fig. 4d and e). This suggests that though metabolic adaptations are vital to bacterial survival following a temperature shift, the pathways involved in such responses will differ depending on the pathogen and their life cycles.

Increased nucleotide biosynthesis has also been identified as a crucial metabolic alteration facilitating the survival of pathogenic bacteria in human blood (60). Some clinical cases of Photorhabdosis have been reported to have become disseminated (2), and the upregulation of the biosynthesis of purines and pyrimidines identified in the metabolic reconstruction could be potentially indicative of a metabolic adaptation facilitating the growth of the bacterium in human blood which contains a lack of nucleotide precursors (60).

Furthermore, nucleotides are the precursors for DNA biosynthesis and replication, and the increased metabolism of both purines and pyrimidines predicted in iEC1073

could contribute to an explanation for the decreased lag growth phase observed in *P. asymbiotica* at 37°C (Fig. 4a). In contrast, at 37°C, the maximum growth rate achieved by the bacterium is less than that observed at 28°C (Fig. 4a). However, looking into the pathways of purine and pyrimidine metabolism in more detail reveals that the reactions and metabolites predicted to be influenced by temperature in iEC1073 are involved in the biosynthesis of DNA precursors (Fig. 4e). These precursors include guanosine triphosphate (gtp), guanosine diphosphate (gdp), and deoxyguanosine (dgsn), for example (Fig. 4e). Such metabolites are necessary for RNA biosynthesis during transcription, potentially suggesting a predicted increase in RNA production at 37°C in *P. asymbiotica*, further supported by the predicted increased net production of uracil at this higher temperature (Fig. 4e). However, this is speculative and would require further experimental investigation to draw any conclusions.

As discussed above, we have predicted a downregulation in reactions involved in oxidative phosphorylation at 37°C (Fig. 4c), which may contribute to the reduced growth rate at higher temperatures. In contrast, most of the genes predicted to be essential in the 37°C reconstruction but not at 28°C are involved in this pathway (File S3). Metabolism of amino acids appears to be downregulated at 37°C compared to 28°C (Fig. 4c) which further supports the theory of a downshift in nutrient utilisation.

As expected, the change in temperature does not have an effect on all metabolic pathways, particularly those which are central to bacterial physiology regardless of the external environment, such as the TCA cycle, as discussed above, and glycolysis (8). Encouragingly, although there are some predicted flux changes in reactions involved in these pathways, predicted flux across these pathways remains relatively unchanged in the 37°C model (Fig. 4d). It is important to note that metabolic pathways integral to the general life cycle of a bacterium will be influenced by transcriptional regulation too, and the integration of transcriptional regulatory networks with the metabolic reconstruction is not performed as part of this study so fluctuations between the model predictions and experimental data is expected.

FBA also has other limitations, and it is important to consider these too. In addition to a lack of transcriptional regulatory information, FBA also does not account for metabolite concentrations or abundance. This type of analysis requires other context-based analysis approaches though the main limiting factor is the current lack of metabolomic data available (49, 61). It can also be assumed that a change in temperature will affect enzyme kinetics, which will undoubtedly have an effect on metabolic reactions and pathways. Constraint-based analysis can be expanded to include reaction kinetics (62) though the vast amount of data required for this is currently unavailable for *Photorhabdus*, or, indeed, most bacterial species.

In conclusion, iEC1073 provides a platform for further investigation of the metabolism of *P. asymbiotica*, and integration with published transcriptomics data has demonstrated the applicability of this model to study the metabolic adaptations the pathogen undergoes in order to survive under stressful environmental conditions concerning temperature but also potentially in response to host switching and changes in the nutritional environment which a mammalian host provides in contrast to an insect host. Integration with further "-omics" data is entirely plausible and could allow further metabolic mechanisms and responses of the pathogen to be elucidated, such as the study of the metabolic adaptations facilitating nematode symbiosis and insect pathogenicity.

## MATERIALS AND METHODS

### Draft reconstruction and initial curation efforts

The genome annotation for *P. asymbiotica* ATCC43949 (1) (*Pa*^ATCC43949^) was downloaded from the National Center for Biotechnology Information (NCBI) (accession number NC_012962.1). The annotated genome was uploaded to the Model SEED (36) (https://

modelseed.org/genomes/) to obtain a draft reconstruction, in XML format and written in Systems Biology Markup Language (SBML). Original Model SEED identifiers were updated to those available in the Biochemical, Genetic, and Genomic (BiGG) models database (37) to ensure universal metabolite and reaction identifiers. In addition, Model SEED gene identifiers were also updated to the locus tags available from the genome annotation.

## BLASTp ortholog search

A BLASTp search was performed using the command line, directed from within R Studio, following the BLAST Command Line Applications User Manual from NCBI (63). Orthologous genes were defined as that with a reciprocal best hit (45).

## Conversion to a mathematical model

The reconstruction was converted to a mathematical model written in Systems Biology Markup Language (SBML) Level 3 Version 1 Core (64) as an XML file. This XML file written in SBML was also converted to a series of Excel files containing model compartments, quantities, reactions, and compounds using SBtab (65) (https://www.sbtab.net/).

## Manual curation

The protocol detailed by Thiele & Palsson was followed (17). The draft reconstruction was extensively manually curated utilizing the literature and several biochemical databases including the Kyoto Encyclopedia of Genes and Genomes (KEGG) pathways database (44, 66, 67) (https://www.genome.jp/kegg/pathway.html), the BiGG knowledge database (37) (http://bigg.ucsd.edu/), and MetaNetX (43, 68–70) (https://www.metanetx.org/) to identify reactions, metabolites, and genes to be added to the model to fill network gaps and reduce the presence of dead-end metabolites. Automated gap filling was performed using the fastGapFill algorithm (71), implemented in the COBRA Toolbox v 3.0 (55) within Matlab (R2019b).

## Constraint-based analysis

All *in silico* experimentation, simulation, and analysis were performed in Matlab (R2019b) using the COBRA Toolbox v 3.0 (55). To predict *in silico* biomass production, the metabolic reconstruction was read into the workspace using the function readCbModel (55) and the biomass reaction (R_BIOMASS) was set as the objective function and optimized through the optimizeCbModel algorithm (55), utilizing FBA. The stoichiometric growth reaction equation was obtained using the printRxnFormula function (55) in the COBRA Toolbox v 3.0 (File S6).

Flux variability analysis (FVA) was utilized to identify whether reactions in the reconstruction were active or inactive. FVA was performed using the fluxVariability algorithm (55). Reactions were deemed as inactive if both the minimum and maximum flux determined by FVA were zero and active if the reaction was carrying flux whereby either the minimum and maximum flux, or both, were non-zero. To identify the number of exchange and transport reactions in iEC1073 and other prokaryotic metabolic reconstructions, the algorithms findExcRxns and findTransRxns (55) were employed, respectively.

## Numerical properties of the reconstruction

In addition to the analysis described during the reconstruction and validation process of the model, further analysis was also performed to investigate the properties of the stoichiometric matrix of the network. Methods for this analysis and results are detailed extensively in File S6 and include the sparsity of the matrix before and after manual gap filling, the nullspace of the matrix and checking the production of biomass precursors. All analyses were performed using the COBRA Toolbox (55).

## Media definition for constraint-based analysis

Constraint-based analysis simulations were performed in two separate media conditions, either with all exchange reactions open to simulate a rich media environment such as growth in LB for example or in a defined minimal medium. A minimal medium supporting the growth of *Photorhabdus* experimentally has been defined (8) and was utilized to simulate constrained growth conditions *in silico*. This medium consists of modified M9 salts, mannose as a carbon source, and casamino acids as a source of nitrogen (Table 1). The uptake rates of trace elements and components of the M9 salts were defined by setting the lower bound to −1,000 mmol/gDw/h, except for oxygen which was set as −2 mmol/gDw/h, each amino acid was set as a percentage of a total uptake rate of −10 mmol/gDw/h depending on the percentage composition of that amino acid in the casein protein structure and the carbon source was set as −10 mmol/gDw/hr.

For experiments utilizing a computational minimal medium, this defined minimal medium was used as a basis and the carbon source adapted to D-glucose (M_glc__D_c) and the nitrogen source to ammonia (M_nh4_c). The experimentally derived media (Table 1) were instrumental during the model reconstruction process, particularly in terms of directing curation efforts. The draft reconstruction obtained from the Model SEED was unable to grow in these conditions, which is suggestive of gaps in the biosynthetic pathways of other metabolites in the reconstruction. To identify the metabolites which could be synthesized by the reconstruction at this stage, all exchange reactions present in the reconstruction were sequentially turned off to identify which further metabolites were required in addition to the media components defined above. This revealed eight primary metabolites which could not yet be synthesized by the model reconstruction and gaps in the pathways involved in this biosynthesis were addressed initially. These metabolites included xanthine (M_xan_c), heme (M_pheme_c), uracil (M_ura_c), glycerol-3-phosphate (M_glyc3p_c), spermidine (M_spmd_c), menaquinone-7 (M_mqn7_c), octadecenoate (M_ocdca_c), and tetradecanoate (M_M02745_c).

## Bacterial strains and metabolic reconstructions used for comparative analysis

For comparative analysis, the genomes of the following five bacterial species were downloaded from NCBI (https://www.ncbi.nlm.nih.gov/); *Escherichia coli* strain K-12 substrain MG1655 (accession number NC_000913.3), *Mycobacterium tuberculosis* H37Rv (accession number NC_000962.3), *Bacillus subtilis* subsp. *subtilis* strain 168 (accession number AL009126.3), *Yersinia pestis* CO92 (accession number NC_003143.1), and *Helicobacter pylori* 26695 (accession number NC_000915.1). The GEMs for these organisms were downloaded from the BiGG database (37). iML1515 for *E. coli* (38) iEK1008 for *M. tuberculosis* (41), iYO844 for *B. subtilis* (39), iPC815 for *Y. pestis* (42), and iIT341 for *H. pylori* (40).

### In silico single-gene knockout experiments

All *in silico* single-gene knockout experiments were performed in unconstrained conditions, with all exchange reactions open to simulate rich media. For gene deletions, the singleGeneDeletion algorithm was implemented in the COBRA Toolbox v 3.0 (55). A gene was considered essential if the rate of biomass production was reduced by >90% compared to the complete model. For comparative analysis with the organisms detailed above, experimental gene essentiality data were downloaded from the Online Gene Essentiality (OGEE) database (53) (https://v3.ogee.info). For data consisting of multiple data sets, the most common essentiality status was used, and for those with no common status, the essentiality status of these genes was considered inconclusive.

### In silico nutrient utilization experiments

Exchange reactions were closed by setting the lower bound to 0 and only those for experimental media components (8) (Table 1) were opened by setting the lower bound

to a non-zero value. For carbon source conditions, the carbon source in the computational media was changed sequentially using casamino acids as the nitrogen source. For nitrogen source conditions, the nitrogen source in the computational media was changed sequentially using mannose as the carbon source. For each tested nutrient source, the corresponding exchange reaction was opened by setting the lower bound to −10 mmol/gDw/h and −1,000 mmol/gDw/h. The model was optimized as described above (55); an increase in biomass production compared to the basal rate was indicative of utilization.

## Integration of transcriptomics data

Differential gene expression data for $Pa^{ATCC43949}$ (8) was integrated into iEC1073 to create context-specific reconstructions representing metabolism of the organism at a temperature of 28°C and at a temperature of 37°C. To create the lower temperature model, flux variability analysis (FVA) was conducted on iEC1073 using the flux-Variability algorithm employed in COBRA Toolbox v 3.0 (55). The upper and lower bounds of all reactions in the network were set to the maximum and minimum flux values, respectively, determined by FVA. The log2fold differential gene expression levels for 37°C compared to 28°C (8) were mapped to each gene in the reconstruction using the findUsedGeneLevels algorithm (55). Expression was subsequently mapped to each reaction in the reconstruction using the selectGenefromGPR algorithm (55); both algorithms were employed in the COBRA Toolbox v 3.0 (55). The selectGenefromGPR algorithm maps expression values to reactions while accounting for "and" and "or" relationships denoted in the GPR association. Reaction bounds were adjusted to represent the expression level of the gene annotated in the gene-protein-reaction association using the formula detailed in Fig. 5, increasing or decreasing the reaction bounds depending on whether the gene associated with the reaction is upregulated or downregulated at a temperature of 37°C compared to 28°C.

Adjusting reaction bounds according to the expression value of the genes in the GPR association means that the resulting flux changes are a direct result of differences in gene expression between the two temperatures. Unlike other context-specific metabolic network reconstruction methods, such as iMAT (72), these temperature-specific models do not contain a subset of active reactions at that particular temperature, but instead include all 2033 reactions described previously but with the reaction bounds adjusting to reflect the experimental transcriptomics data (8).

*In silico* single-gene knockouts were repeated with the two temperature-dependent models following the method described above.

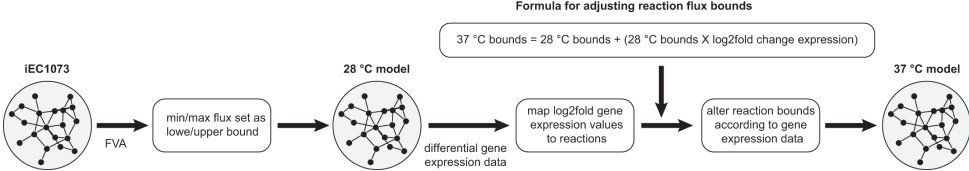

**Formula for adjusting reaction flux bounds**

37 °C bounds = 28 °C bounds + (28 °C bounds X log2fold change expression)

**FIG 5** Adjusting reaction bounds in iEC1073 to create temperature-specific reconstructions. Integrating gene expression data (8) for growth of *P. asymbiotica* at 28°C and 37°C to create temperature-specific networks. Flux variability analysis (FVA) was performed on iEC1073 and the upper and lower bounds of each reaction in the reconstruction set as the maximum and minimum flux determined by FVA, respectively, to create the 28°C model. This model was used as a template to create the 37°C model. Log2fold gene expression changes at 37°C compared to 28°C (8) were mapped to each reaction in the network, accounting for the gene-protein-reaction association. The bounds of reactions in the template 28°C model were adjusted according to this mapped log2fold expression change to create the 37°C model. All analyses were performed in the COBRA Toolbox v 3.0 (55) implemented in Matlab (R2019b).

## Visualization of temperature-specific alterations in the metabolic network

For each temperature-dependent reconstruction, the overall production of a particular metabolite was calculated by adding the predicted flux of all associated reactions, where the particular metabolite is a product. The ratio of overall metabolite production at 37°C compared to 28°C was calculated, and the metabolite was considered upregulated or downregulated if the ratio is >1.05 or <0.95 (increased or decreased by a threshold of 5%), respectively. The total number of reactions which were changed between the two temperature-specific models was determined, and a hypergeometric statistical test was applied to perform enrichment analysis for the upregulated and downregulated reactions in each pathway to calculate $P$-values. In order to visualize these temperature-induced changes in the network, first, the reactions where at least one metabolite is changed at 37°C were identified. Fluxes of two identical reactions were added to simplify the network. Some of the highly connected hub metabolites including h2o, h, nadph, nadp, nadh, nad, pi, atp, and adp were removed from the network for better visualization. The network sub clusters were created using the cluster_fast_greedy function from the igraph package in R (73).

## Experimental growth curves of *P. asymbiotica*

$Pa^{ATCC43949}$ was streaked onto LB agar and left to incubate at 28°C and 37°C for 48 h. Overnight cultures were prepared in triplicate in 10 mL LB and left to incubate at 28°C and 37°C, shaking at ~200 rpm. Overnight cultures were diluted 1:50 in 10 mL fresh LB and left to grow to an OD of 0.4–0.6. Cultures were subsequently diluted to on OD of 0.01 and 100 µL aliquots added to a 96-well plate. Growth of cultures was measured at 28°C and 37°C over 48 h using a FLUOStar Omega Plate Reader, measuring OD every 15 min.

Growth of $Pa^{ATCC43949}$ was also assessed in minimal media (File S5; Fig. S4). The same protocol as above was followed for these growths though preparing cultures in LB and minimal media. Minimal media were prepared following the media detailed by Mulley et al. (8) using a modified M9 salt buffer at pH 6.8 (12.5 mM $Na_2HPO_4$, 8.5 mM NaCl, 22 mM $KH_2PO_4$, 2 mM $MgSO_4$, and 100 µM $CaCl_2$) (8), 20 mM glucose, 0.05% (wt/vol) casamino acids and a trace element solution modified from Helmholtz Zentrum München (13.4 mM EDTA, 3.1 mM $FeCl_3\cdot6H_2O$, 0.62 mM $ZnCl_2$, 8.1 µM $MnCl_2\cdot4H_2O$, 76 µM $CuCl_2\cdot2H_2O$ and 42 µM $CoCl_2\cdot2H_2O$). Growth was measured at 28°C. All data were analyzed using GraphPad Prism v 9.0.

## AUTHOR AFFILIATIONS

[1]Warwick Medical School, University of Warwick, Gibbet Hill Campus, Coventry, United Kingdom
[2]Bioinformatics Research Technology Platform, University of Warwick, Warwick, United Kingdom
[3]Department of Biology, College of Science, United Arab Emirates University, Al-Ain, United Arab Emirates

## AUTHOR ORCIDs

Elena Lucy Carter ⓘ http://orcid.org/0000-0003-0030-1527
Chrystala Constantinidou ⓘ http://orcid.org/0000-0001-7210-3816
Mohammad Tauqeer Alam ⓘ http://orcid.org/0000-0002-6872-0691

## FUNDING

| Funder | Grant(s) | Author(s) |
| --- | --- | --- |
| United Arab Emirates University (UAEU) | G00003688, G00004152 | Mohammad Tauqeer Alam |

| Funder | Grant(s) | Author(s) |
|---|---|---|
| MRC Doctoral Training Partnership at the University of Warwick | MR/N014294/1 | Elena Lucy Carter |

## AUTHOR CONTRIBUTIONS

Elena Lucy Carter, Formal analysis, Investigation, Methodology, Writing – original draft | Nicholas R. Waterfield, Funding acquisition, Supervision, Writing – review and editing | Chrystala Constantinidou, Supervision, Writing – review and editing | Mohammad Tauqeer Alam, Conceptualization, Formal analysis, Funding acquisition, Supervision, Writing – original draft, Writing – review and editing

## DATA AVAILABILITY

The genome-scale metabolic model of P. asymbiotica, iEC1073, has been uploaded to BioModels (https://www.ebi.ac.uk/biomodels/) with the ID MODEL2309110001. The model is written in SBML L3V1 and available as an XML file.

## ADDITIONAL FILES

The following material is available online.

### Supplemental Material

**File S1 (mSystems00970-23-S0001.docx).** BLASTp ortholog search.
**File S2 (mSystems00970-23-S0002.xlsx).** iEC1073 model information.
**File S3 (mSystems00970-23-S0003.xlsx).** Gene-knockout results and concordance comparison.
**File S4 (mSystems00970-23-S0004.xlsx).** Nutrient source utilization by iEC1073 compared to Biolog phenotype microarray data.
**File S5 (mSystems00970-23-S0005.xlsx).** Temperature-specific reconstructions.
**File S6 (mSystems00970-23-S0006.docx).** Further stoichiometric matrix analysis.
**Supplemental Figures (mSystems00970-23-S0007.docx).** Figures S1-S4.

### Open Peer Review

**PEER REVIEW HISTORY (review-history.pdf).** An accounting of the reviewer comments and feedback.

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
