## [Reviewer comments · mSystems]

A Temperature-Induced Metabolic Shift in the Emerging Human Pathogen *Photorhabdus asymbiotica*

Mohammad Alam, Elena Carter, Nicholas Waterfield, and Chrystala Constantinidou

Corresponding Author(s): Mohammad Alam, United Arab Emirates University

Review Timeline:

Submission Date:

September 19, 2023

Accepted:

November 29, 2023

Editor: Jon Sanders

Reviewer(s): Disclosure of reviewer identity is with reference to reviewer comments included in decision letter(s). The following individuals involved in review of your submission have agreed to reveal their identity: Maksim Zakhartsev (Reviewer #2)

Transaction Report:

DOI: <https://doi.org/10.1128/msystems.00970-23>

Re: mSystems00970-23 (**A Temperature-Induced Metabolic Shift in the Emerging Human Pathogen *Photorhabdus asymbiotica***)

Dear Dr. Mohammad Tauqeer Alam:

Thank you for resubmission of this manuscript. As you will see below, both reviewers found the revisions to substantially improve the paper, and recommended to accept. Reviewer 2 did note that it would be helpful to include a brief discussion regarding the choice of strain used, which I agree with as well. I would also suggest slightly modifying the legends in Figure 4 to indicate under which condition pathways were upregulated or downregulated (i.e. up *at 37{degree sign}*).

Your manuscript has been accepted, and I am forwarding it to the ASM production staff for publication. Your paper will first be checked to make sure all elements meet the technical requirements. ASM staff will contact you if anything needs to be revised before copyediting and production can begin. Otherwise, you will be notified when your proofs are ready to be viewed.

Featured Image Submissions: If you would like to submit a potential Featured Image, please email a file and a short legend to mSystems@asmusa.org. Please note that we can only consider images that (i) the authors created or own and (ii) have not been previously published. By submitting, you agree that the image can be used under the same terms as the published article. File requirements: square dimensions (4" x 4"), 300 dpi resolution, RGB colorspace, TIF file format.

Sincerely,

Jon Sanders
Editor
mSystems

Reviewer #1 (Comments for the Author):

The authors have taken considerable effort to put together a very good response, very carefully done. Essentially all my concerns have been addressed.

At this stage, while only time will tell if this is useful, the study itself is well done and quite complete for a first paper. The clarifications on temperature and flux assumptions now make the paper more precise.

Reviewer #2 (Comments for the Author):

Phototransducing bacterium ATCC43949 has been isolated in 1977 from the blood of an 80-year-old female patient with endocarditis, in Maryland, USA.

- https://www.genome.jp/kegg-bin/show_organism?org=pay
- <https://www.atcc.org/products/43949>

The annotated genome *Phototransducing bacterium* ATCC43949 has been downloaded from NCBI (lines 556-557). Thus, the strain that has been used for model reconstruction was isolated from 37°C environment, i.e. it already is the pathogenic strain. Then the ATCC43949 strain has been grown at 28°C and 37°C for comparative purposes (lines 734-739). Then a methodological question arises - why authors have not used a wild strain as a reference for the comparison 28°C versus 37°C in order to reveal temperature induced adaptation of strain ATCC43949? It seems that such approach would be more informative for the purposes of revealing the temperature induced metabolic adaptation of the pathogenic strain that allows survival at 37°C. As a consequence, the authors have identified significant difference related only to upregulation of following pathways: purine metabolism and synthesis of (p)Gpp. On my point of view, the found difference does not allow explicit explanation why ATCC43949 is able to grow at 37°C, while wild *Phototransducing bacterium* species unable to grow above 34°C. In the same time, authors explicitly stated several times that switch from a pathogenic to mutualistic life cycle in *Phototransducing bacterium* species is characterized by the production of antibiotics (stilbene), pigments and bioluminescence. Thus, these metabolic features could have been used as "metabolic load" in the reconstructed model. On my request, the authors have added the complete stoichiometric growth equation (Supplementary File 6), from which follows that stilbene is not in the list of the end-products, and photon is a reactant (not a product) of the equation.

On my point of view, the strength of FBA significantly increases when used the experimentally measured extracellular in/output metabolic fluxes (i.e. 28°C versus 37°C) [including bioluminescence] in order to constrain the model. However, the authors have stated that experimentally measured metabolic fluxes were "outside the scope of this study", thus giving them space for speculative conclusions.

Also I would recommend the authors to check the model quality using MEMOTE tool.

Photorhabdus asymbiotica ATCC43949 has been isolated in 1977 from the blood of an 80-year-old female patient with endocarditis, in Maryland, USA.

- https://www.genome.jp/kegg-bin/show_organism?org=pay
- <https://www.atcc.org/products/43949>

The annotated genome *Photorhabdus asymbiotica* ATCC43949 has been downloaded from NCBI (lines 556-557). Thus, the strain that has been used for model reconstruction was isolated from 37°C environment, i.e. it already is the pathogenic strain. Then the ATCC43949 strain has been grown at 28°C and 37°C for comparative purposes (lines 734-739). Then a methodological question arises – why authors have not used a wild strain as a reference for the comparison 28°C versus 37°C in order to reveal temperature induced adaptation of strain ATCC43949? It seems that such approach would be more informative for the purposes of revealing the temperature induced metabolic adaptation of the pathogenic strain that allows survival at 37°C. As a consequence, the authors have identified significant difference related only to upregulation of following pathways: purine metabolism and synthesis of (p)ppGpp. On my point of view, the found difference does not allow explicit explanation why ATCC43949 is able to grow at 37°C, while wild *Photorhabdus* species unable to grow above 34°C. In the same time, authors explicitly stated several times that switch from a pathogenic to mutualistic life cycle in *Photorhabdus* species is characterized by the production of antibiotics (stilbene), pigments and bioluminescence. Thus, these metabolic features could have been used as “metabolic load” in the reconstructed model. On my request, the authors have added the complete stoichiometric growth equation (Supplementary File 6), from which follows that stilbene is not in the list of the end-products, and photon is a reactant (not a product) of the equation.

On my point of view, the strength of FBA significantly increases when used the experimentally measured extracellular in/output metabolic fluxes (i.e. 28°C versus 37°C) [including bioluminescence] in order to constrain the model. However, the authors have stated that experimentally measured metabolic fluxes were “outside the scope of this study”, thus giving them space for speculative conclusions.

Also I would recommend the authors to check the model quality using MEMOTEtool.